# From Microbial Consortia to Ecosystem Resilience: The Integrative Roles of Holobionts in Stress Biology

**DOI:** 10.3390/biology14091203

**Published:** 2025-09-06

**Authors:** Maximino Manzanera

**Affiliations:** Institute for Water Research and Department of Microbiology, University of Granada, CL Ramón y Cajal No. 4, 18071 Granada, Spain; manzanera@ugr.es; Tel.: +34-958-248324; Fax: +34-958-243094

**Keywords:** holobiont, microbiome-host coevolution, stress adaptation, drought stress, methane oxidizing bacteria, multi-organ axes

## Abstract

The holobiont framework, encompassing host organisms and their microbial symbionts as integrated biological systems, has revolutionized perspectives on environmental stress responses across diverse taxa. This review synthesizes evidence that holobionts, from plants to animals, rely on microbial partnerships to thrive under environmental challenges. We trace the evolutionary origins of these alliances to ancient microbial consortia, which enabled metabolic innovation and stress resilience. In plants, microbiome-mediated mechanisms (e.g., rhizosphere methanotrophs enhancing drought tolerance) optimize nutrient acquisition and pathogen defense. In animals, the gut microbiota influences systemic health through multi-organ axes (e.g., gut–brain communication via microbial metabolites). Cross-kingdom parallels reveal conserved principles, such as volatile organic compounds (VOCs) facilitating inter-holobiont signaling. Emerging tools, like microbiome engineering, harness these interactions for climate-resilient agriculture and precision medicine. By framing adaptation as a collective trait of the holobiont, this work bridges evolutionary biology, microbiology, and ecology, offering actionable insights for sustainability and health.

## 1. Introduction

Many studies indicate that life on Earth emerged in a microbial world, where prokaryotic alliances shaped the evolutionary trajectory of multicellular organisms. The holobiont theoretical framework proposes that host organisms and their resident microbial communities undergo reciprocal evolutionary processes as mutually dependent assemblages, wherein microbial influences span from molecular mechanisms to landscape-level ecological dynamics [1,2]. While previous reviews have examined specific aspects of host–microbiome interactions, this article uniquely integrates three underexplored dimensions. Firstly, we explore the evolutionary foundations, as we trace how ancient microbial consortia (protomicrobiomes) enabled the transition to multicellularity by facilitating metabolic innovation and stress partitioning—a legacy preserved in modern holobionts. Secondly, we establish mechanistic cross-kingdom parallels, as we juxtapose microbiome-mediated stress responses in plants (e.g., rhizosphere methanotrophs alleviating drought) and animals (e.g., gut microbiota modulating the hypothalamic–pituitary–adrenal axis or the HPA axis), identifying conserved signaling principles (e.g., redox metabolites, neuroactive compounds). Finally, we focus on ecological networks, as we expand the holobiont concept beyond individual hosts to inter-holobiont interactions mediated by microbial volatiles and “bacterial highways,” illustrating how microbial dispersal shapes ecosystem resilience. For instance, marine holobionts like coral–algae symbioses (e.g., *Symbiodinium* spp. in reef-building corals) demonstrate how microbiome shifts under thermal stress correlate with bleaching events, as shown by metatranscriptomic profiling [3]. Similarly, extremophile holobionts, such as hydrothermal vent tubeworms (*Riftia pachyptila*) reliant on chemosynthetic *Endoriftia* bacteria, exemplify metabolic innovation in high-pressure, anoxic environments [4]. These systems underscore the need to expand beyond terrestrial models.

This synthesis challenges the traditional host-centric view of adaptation, arguing that stress responses are emergent properties of the holobiont. By unifying data from molecular microbiology, systems biology, and ecology, we provide a framework to harness microbiome plasticity for climate-resilient agriculture and precision medicine. Our analysis draws on recent advances in metagenomics, metabolomics, and experimental evolution, with an emphasis on recently published studies.

While excellent reviews have examined holobiont evolution (e.g., Bordenstein and Theis, 2015 [2]), molecular mechanisms [5,6], or ecological roles [7] in isolation, this work uniquely integrates these strands. We argue that only by bridging deep-time symbiosis, cross-kingdom stress-response principles, and inter-holobiont network dynamics can we predict adaptation in rapidly changing environments—an imperative for addressing climate and biomedical challenges.

## 2. From Primordial Soup to Coevolved Consortia: The Deep Roots of Symbiosis

Contemporary paleobiological data indicates accelerated biogenesis on early Earth, with prokaryotic dominance established approximately 3.5–3.7 billion years before present. Eukaryotic lineages are theorized to have emerged subsequently (~2.1 billion years ago) through endosymbiotic incorporation events in which ancestral archaea may have engulfed prokaryotes, giving rise to mitochondria and chloroplasts [8]. The subsequent evolution of multicellularity (~0.6–0.8 By) is thought to have occurred within microbe-rich environments, where symbiotic partnerships may have played a key role in shaping holobionts (host–microbiome units) as a potential framework for eukaryotic complexity [1]. These symbioses might have contributed to metabolic innovation, stress adaptation mechanisms, and developmental cues, with microbial interactions possibly influencing processes such as terrestrial colonization (e.g., through plant–mycorrhizal networks) and cellular specialization [9,10].

On the early Earth—an anoxic world inhabited exclusively by microorganisms—a constantly shifting balance of environmental conditions drove the evolution of highly adaptable metabolic strategies. Atmospheric oxygen remained below 0.001% for nearly half of Earth’s history, driving the emergence of prokaryotes that exploited inorganic compounds for energy and fixed CO_2_ as a carbon source, such as methanogens [11]. Other microbes, including purple and green sulfur bacteria, evolved to utilize H_2_S and reduced compounds as electron donors [12,13]. Fermentative anaerobes also appeared [14], alongside metabolisms employing alternative terminal electron acceptors such as NO_3_^−^, SO_4_^2−^, and Fe^3+^.

In early microbial ecosystems, both competitive and symbiotic interactions shaped community dynamics. These interactions fostered the formation of metabolically integrated microbial consortia characterized by the active exchange of metabolites and genetic material—interdependent communities where microorganisms exchange metabolites and genetic material to optimize collective survival. For example, in anaerobic environments, syntrophic bacteria and methanogenic archaea cooperate tightly: bacteria ferment organic compounds, producing hydrogen and acetate, while archaea consume these byproducts to generate methane, preventing metabolic inhibition. Such consortia, which we designate as protomicrobiomes, represent the first functional microbiomes. Specialization emerged not only at the level of individual microorganisms but also at the community level, leading to the development of functionally specialized microbiomes. Such communities are hypothesized to have preceded and facilitated the cellular and tissue specialization seen in multicellular life. The advent of multicellular organisms coincided with the appearance of their associated microbiomes, forming what we call protoholobionts, precursors of holobionts. These protoholobionts represent an evolutionary transition toward complex holobionts—multicellular eukaryotes forming obligate partnerships with phylogenetically diverse microbiota (bacteria, archaea, fungi, viruses, and protists)—where host–microbiome metabolic complementarity established a functionally integrated metaorganism with emergent ecological capabilities [2,15] (Figure 1).

Marine environments provide compelling examples of ancient protomicrobiome dynamics that persist today. Deep-sea hydrothermal vent communities exemplify these primordial partnerships, where sulfur-oxidizing bacteria (e.g., *Thiomicrospira* spp.) form obligate associations with tube worms (*Riftia pachyptila*), creating holobionts capable of thriving in extreme temperature gradients (400 °C) and heavy metal concentrations that would be lethal to either partner alone [16,17]. Similarly, the marine sponge *Aplysina aerophoba* hosts over 40 distinct bacterial phyla, including extremophilic archaea that contribute ammonia oxidation and specialized secondary metabolite production, demonstrating how complex microbial consortia enable host survival in oligotrophic marine environments [18,19].

A sustained rise in atmospheric oxygen levels occurred between 2.1 and 2.4 billion years ago, marking the Great Oxidation Event (GOE) [12,20]. This global shift in environmental conditions also impacted holobionts, beginning with adaptations in their associated microbiomes to enhance the survival of the multicellular eukaryotic host. Failure to adapt resulted in extinction. According to evolutionary theory, adaptability is key to survival, and microbiome modification—via both vertical and horizontal gene transfer—offers a faster and more flexible mechanism than genomic changes in the host.

Surprisingly, despite the similar number of prokaryotic and eukaryotic cells in multicellular organisms such as humans, the functional genetic contribution from prokaryotes far exceeds that of the host, with an estimated 3–10 million microbial genes compared to approximately 20,000 human genes. This vast repertoire of prokaryotic genes provides a dynamic source of proteins that contribute significantly to holobiont physiology. The evolutionary plasticity of microbial genomes, facilitated by horizontal gene transfer and alternative regulatory mechanisms, enables rapid metabolic adjustments in response to environmental change. For example, the evolutionary plasticity of microbial genomes, facilitated by horizontal gene transfer and alternative regulatory mechanisms, enables rapid metabolic adjustments in response to environmental perturbations. These adaptive responses are mediated through diverse microbial metabolites that serve as crucial signaling molecules. Notably, intestinal microbe-synthesized short-chain fatty acids (SCFAs)—encompassing acetate, propionate, and butyrate—demonstrate the capacity for regulating the host metabolic homeostasis and immunological responses across timeframes spanning minutes to hours [21].

Similarly, tryptophan metabolism by commensal bacteria yields neuroactive compounds such as serotonin and indole derivatives, which directly influence enteric nervous system activity and central neurological processes [22]. The microbial production of γ-aminobutyric acid (GABA) contributes to the regulation of neuronal excitability and stress responses, while bacterial quorum-sensing molecules like acyl-homoserine lactones facilitate population-level coordination of gene expression during environmental challenges [23]. These microbial metabolites collectively form an intricate biochemical communication network that translates environmental signals into physiological responses, exemplifying the holobiont’s capacity for rapid adaptation [24]. Therefore, to achieve long-term adaptations, the holobiont can implement genetic modifications at the eukaryotic level, such as the epigenetic reprogramming of stress-response genes (e.g., DNA methylation in the promoter regions of heat shock proteins), somatic mutations in immune-related genes (e.g., immunoglobulin gene rearrangements), or selection for advantageous alleles (e.g., lactase persistence variants) [25]. However, in response to rapidly changing conditions like infectious disease challenges, the microbiome demonstrates remarkable plasticity through several mechanisms, including the rapid horizontal gene transfer of virulence factors or antibiotic resistance genes among commensals, the compositional shifts favoring protective taxa (e.g., increased abundance of Bacteroidetes during enteric infections), and the production of antimicrobial peptides (e.g., bacteriocins) that competitively exclude pathogens. This microbial adaptability provides immediate protection, while eukaryotic genetic systems enact slower evolutionary responses [26].

## 3. The Plant Microbiome

Within plant–microbial associations, compartmentalized microbial communities inhabit distinct anatomical niches, with root-zone assemblages representing the most extensively characterized system. This rhizospheric habitat offers conducive environmental parameters—including hydration gradients, oxygen availability, and exudate-derived carbon sources—facilitating intensive microbial establishment. Other microbiomes include the endosphere (within roots), phyllosphere (aerial parts), anthosphere (flowers), and spermosphere (seeds), each performing specialized functions.

The rhizosphere microbiome plays key roles in mitigating abiotic stresses (e.g., drought, salinity) and biotic stresses (e.g., pathogen defense) and enhancing nutrient acquisition (e.g., nitrogen metabolism, phosphate and potassium solubilization) [27,28,29]. The rhizosphere microbiome comprises complex ecological relationships, ranging from mutualism to commensalism and conditional pathogenicity. While certain microorganisms engage in mutualistic interactions—such as mycorrhizal fungi enhancing phosphate solubilization (obligate mutualists) or nitrogen-fixing rhizobia exchanging nutrients with host plants (facultative mutualists)—many exhibit context-dependent behaviors [30]. For example, the typically beneficial *Pseudomonas* strains can become pathogenic under host stress, and the commensal *Enterobacter* species may enhance drought tolerance without directly reciprocating benefits. These interaction types, ranging from obligate mutualism to conditional pathogenicity, are summarized in Table 1, together with representative examples, mechanisms, and outcomes.

Microbial cooperation emerges through cross-feeding synergies (e.g., auxin-producing bacteria stimulating fungal growth), collective biofilm formation mediated by quorum sensing, and coordinated suppression of pathogens via antibiotic production or niche competition, among others [24,32]. These dynamic interactions collectively enhance plant resilience to abiotic (e.g., drought, salinity) and biotic (e.g., pathogen defense) stresses while optimizing nutrient acquisition through complementary metabolic functions (e.g., nitrogen fixation coupled with phosphate solubilization) [32,35].

In contrast to the rhizosphere, the anthosphere microbiome plays a central role in reproductive fitness by influencing floral physiology and ecology [36,37]. Microbial communities colonizing flowers can shape volatile organic compound (VOC) profiles, thereby enhancing pollinator attraction and modulating the specificity of plant–pollinator interactions [38]. Certain yeasts and bacteria metabolize nectar sugars to produce ethanol, organic acids, and distinctive aromatic compounds that serve as olfactory cues for pollinators [38]. At the same time, floral-associated microbes contribute to pathogen defense by producing antimicrobial metabolites, outcompeting invasive fungi and bacteria, and priming local plant immune responses. Collectively, these functions extend beyond flower-level protection to regulate reproductive success, pollen dispersal efficiency, and, ultimately, the development of the entire holobiont [39]. These microbiome specializations influence and modify local environments to favor holobiont development. For instance, microbial activity can alter soil pH to enhance plant growth. In more extreme scenarios, such as arid methane-rich environments, methanotrophs oxidize CH_4_ into CO_2_ and H_2_O, improving water and carbon availability for plant metabolism [31]. This mutualistic environmental engineering enhances holobiont survival and microbial persistence, underscoring the functional importance of microbiome-driven ecological adaptation.

While bacterial communities can persist independently of plants in soil ecosystems, the rapid formation of functional holobionts occurs through chemotactic recruitment toward plant exudates, a process mediated by conserved molecular signaling. For instance, specific flavonoids (e.g., luteolin) and dicarboxylic acids (e.g., malate) serve as chemoattractants that guide microbial assembly within hours of seed germination [25].

The success of microbiome metabolic pathways is evolutionarily encoded through feedback loops. On the one hand, microbial byproducts (e.g., auxins, 1-aminocyclopropane-1-carboxylate deaminase, or ACC deaminase) directly enhance host fitness, creating selective pressure for mutualism. On the other hand, eukaryotic hosts actively reward beneficial symbionts through preferential carbon allocation (e.g., increased sucrose secretion for nitrogen-fixing rhizobia); in addition, horizontal gene transfer disseminates adaptive traits (e.g., nodulation genes) across microbial populations, as demonstrated in experimental rhizosphere evolution studies [40]. This dynamic reciprocity ensures that microbiome functions aligning with host needs are amplified over evolutionary timescales.

Extremophilic plant holobionts provide additional evidence of microbiome-mediated environmental adaptation. *Spartina alterniflora* (salt marsh grass) partners with halotolerant endophytes such as *Halomonas* and *Marinobacter* species that not only survive, but actively remediate salt concentrations exceeding 35‰ through compatible solute production and Na^+^/K^+^ efflux mechanisms [41]. These bacterial partners produce osmoprotectants (glycine betaine, proline) that are directly utilized by plant cells, while simultaneously sequestering toxic metals through metallothionein-like compounds. In arctic tundra, *Oxyria digyna* forms partnerships with psychrotrophic bacteria, including *Pseudomonas* and *Flavobacterium* strains that remain metabolically active at −15 °C through antifreeze protein production and membrane fluidity modifications, extending the plant’s growing season by up to 30 days annually [42,43].

## 4. The Animal Microbiome: Establishment, Composition, and Host Interactions

Animal microbiome assembly occurs via three principal acquisition pathways: (1) maternal–offspring transfer encompassing parturition canal exposure, lactational inoculation, and dermal contact (e.g., transfer of *Bifidobacterium longum* during vaginal delivery and nursing); (2) environmental recruitment from atmospheric, aquatic, edaphic, and nutritional sources (e.g., Bacteroides species from plant fiber consumption); and (3) horizontal transfer through behavioral interactions like grooming and sharing habitats (e.g., *Faecalibacterium prausnitzii* spreading among social mammals) [44,45,46].

There are arguments to consider the presence of niche-specific microbial communities. In this regard, gut microbiota is dominated by Firmicutes (*Lactobacillus, Ruminococcus*), Bacteroidetes (*Bacteroides, Prevotella*), and Actinobacteria (*Bifidobacterium*), with archaeal methanogens (*Methanobrevibacter smithii*) contributing to energy harvesting [47]. Meanwhile, skin communities show regional specialization: sebaceous zones harbor *Cutibacterium acnes* and *Staphylococcus epidermidis*, while moist areas are enriched with *Corynebacterium* and *Malassezia* species [48]. On the other hand, the respiratory tract maintains distinct populations of *Streptococcus pneumoniae* and *Moraxella catarrhalis* in nasal passages [49].

It is important to consider the host–microbiome co-adaptation, where host genetics significantly shape microbial composition through mechanisms like the HLA-mediated selection of *Bacteroides* strains and the diet-driven enrichment of bile-tolerant *Alistipes* species [50]. Concurrently, the microbiome influences host development by promoting gut-associated lymphoid tissue formation through *Clostridia*-induced regulatory T cell differentiation and modulating behavior via *Lactobacillus rhamnosus*-mediated GABA receptor expression [51]. This bidirectional interaction creates a co-evolutionary feedback loop where microbial functions (e.g., nutrient metabolism, immune education) and host physiology (e.g., digestive efficiency, mucosal immunity) become mutually optimized.

The human microbiome, which is particularly dense and diverse in the gut, constitutes a complex microbial ecosystem with critical implications for host health. Its development begins at birth and stabilizes into an individual-specific composition, although consistent patterns allow classification into enterotypes. Microbial homeostasis, or eubiosis, supports systemic equilibrium, while dysbiosis has been associated with inflammatory, metabolic, and neurological disorders.

Intestinal microbial communities engage in reciprocal communication with central nervous structures through neural, immunological, and hormonal networks, regulating neurotransmitter synthesis, enteric barrier integrity, and hypothalamic–pituitary–adrenal system activity. This gut–brain axis influences mood, cognition, and the pathogenesis of diseases such as depression, autism, and Alzheimer’s disease [22]. Beyond the gut–brain axis, animal microbiomes engage in multi-organ crosstalk through distinct bidirectional axes, including the gut–liver axis, where microbial metabolites (e.g., secondary bile acids) modulate hepatic detoxification and inflammation via portal circulation; the gut–lung axis, mediated by SCFA-primed immune cells trafficking to respiratory mucosa; the gut–skin axis, driven by tryptophan metabolite regulation of cutaneous immunity; and the cardiovascular axis, wherein bacterial TMAO production influences endothelial function. These axes collectively demonstrate how spatially segregated microbiomes systemically coordinate host physiology through metabolite diffusion, immune cell trafficking, and neuroendocrine signaling [52].

Lymph nodes are strategically located near microbe-rich sites (e.g., mesenteric, cervical, inguinal, and axillary regions) and are traditionally linked to immune filtration and lipid transport. However, emerging evidence suggests they play a broader role in distributing microbial metabolites to less colonized tissues via the lymphatic system, potentially regulating host metabolism and systemic homeostasis.

Studies of the gut–liver axis have demonstrated that metabolite transport occurs via both mesenteric lymphatics and venous circulation, modulating hepatic function [34,53]. This observation supports the possibility of analogous axes in other lymph node-dense, microbe-rich regions, such as the axillary, cervical, inguinal area, where lymph composition could affect both host cell metabolism and microbiome structure.

The cervical and axillary regions are key anatomical sites where dense microbial colonization interfaces with regional lymph nodes. These spatially organized interfaces between microbial reservoirs and regional lymph nodes are summarized in Table 2, together with their putative mechanisms and systemic implications. In the cervical region, microbial hotspots include the tonsillar crypts (dominated by anaerobic genera like *Fusobacterium* and *Prevotella*), nasopharyngeal mucosa (colonized by *Streptococcus pneumoniae* and *Neisseria* spp.), and salivary glands (hosting oral streptococci and *Veillonella*). The axillary region harbors distinct communities in apocrine sweat glands (*Corynebacterium* and *Staphylococcus epidermidis* metabolizing sweat components), hair follicles (*Cutibacterium acnes* and *Malassezia* spp.), and sebaceous glands (*Staphylococcus hominis* and *Aerococcus*) [54].

These microbial reservoirs actively engage with local lymphoid tissues. Cervical lymph nodes sample oral and nasal microbiota to shape mucosal immunoglobulin A (IgA) responses, while axillary lymph nodes process skin-derived antigens to regulate cutaneous immunity against pathogens like *Staphylococcus aureus* [55,56]. This spatial organization of microbial–lymphatic interfaces enables localized immune surveillance while also contributing to the maintenance of the systemic host–microbiome homeostasis.

While most research emphasizes neural pathways—particularly in the highly innervated mesenteric region—gut microbes also produce neuroactive compounds capable of engaging both the enteric and central nervous systems. These findings reinforce the microbiome’s central role in host physiology across multiple organ systems [34,53].

## 5. On the Interaction Between Different Holobionts

While microbial metabolites such as short-chain fatty acids (SCFAs) and volatile organic compounds (VOCs) appear to function as signaling molecules across diverse biological systems, claims of their universality require careful scrutiny. The assumption that these compounds operate through conserved mechanisms across kingdoms oversimplifies the complex, context-dependent nature of host–microbiome interactions and risks creating misleading generalizations about holobiont communication networks.

Cross-species microbial transfer occurs through multiple ecological routes: (1) plant–to–animal transmission via the consumption of phyllosphere Methylobacterium or rhizosphere *Pseudomonas*; (2) animal–to–animal exchange of *Akkermansia muciniphila* through coprophagy, or *Staphylococcus hominis* via direct contact [57,58]. However, successful establishment is highly constrained by host-specific factors that challenge universal signaling paradigms. For instance, while *Escherichia coli* demonstrates apparent generalist capabilities across mammalian hosts, its metabolite production varies dramatically depending on host diet, gut pH, and competing microbiota—factors that can completely alter downstream signaling outcomes [59]. Figure 2 presents some of these ecological routes.

Hosts actively shape microbial communities through ecological filtering (avian crop pH selection for acid-tolerant *Lactobacillus*) and behavioral adaptations (primate geophagy for detoxifying *Bacillus*) [60,61]. Conversely, microbes modify host niches through mucus induction (*Vibrio shiloi* in corals) and direct manipulation of host signaling (*Helicobacter pylori* CagA-mediated gastric acid regulation) [62,63]. These interactions drive evolutionary outcomes such as host–microbe co-speciation (parallel divergence of gut microbiomes and hominid lineages) and local adaptation (geographic *Bifidobacterium* strains matching lactase persistence genotypes) [64].

Microbial assemblages affiliated with diverse organisms—spanning human, animal, and plant systems—exert physiological influence on target holobionts via VOC synthesis. These microbially derived gaseous signals function at spatial distances, affecting both symbiotic hosts and proximate organisms. In plants, certain microbial VOCs enhance stress tolerance (e.g., drought or salinity) and facilitate nutrient acquisition, such as iron or sulfur uptake under limiting conditions [65]. These gaseous signals are not restricted to the host but can affect the surrounding plants as well.

Similar inter-organismal VOC-mediated communication occurs in humans and non-human animals. Microbial communities produce gaseous metabolites that influence not only their host’s physiology and behavior, but also that of nearby individuals [66]. Since these emissions are shaped by the dynamic composition of the microbiome—which varies with age, hormonal state, and psychological condition—they serve as indicators of developmental stage and emotional state, potentially reflecting traits such as trustworthiness or aggression. Microbiomes within the human body, particularly the gut and oral microbiota, also produce volatile organic compounds (VOCs) that influence host physiology. These VOCs include short-chain fatty acids (e.g., acetate, propionate, butyrate), sulfur-containing compounds, amines, and phenolic derivatives, which can modulate systemic inflammation, neurotransmission, and metabolic regulation. For example, butyrate produced by gut bacteria promotes intestinal barrier integrity and has anti-inflammatory effects [67], while trimethylamine (TMA), derived from the microbial metabolism of choline, is converted in the liver to trimethylamine N-oxide (TMAO), a molecule associated with cardiovascular disease risk [68]. Moreover, certain gut-derived VOCs, including indole and *p*-cresol, are neuroactive and have been implicated in modulating behavior and neurological function [69]. These findings underscore that VOCs produced endogenously by the microbiota serve as functional mediators in microbiome–host communication.

However, the interpretation of VOCs as universal signaling molecules is problematic when examined critically. In plants, certain microbial VOCs have been reported to enhance stress tolerance and facilitate nutrient acquisition [65]. Furthermore, studies often fail to account for dose-dependent responses, temporal variability, and the presence of confounding environmental factors. For example, the same bacterial VOCs that enhance drought tolerance in Arabidopsis at low concentrations can become phytotoxic at higher concentrations, while their effects are completely negated in the presence of competing root exudates from neighboring plants [70]. This concentration dependency and context specificity directly contradict the notion of a universal signaling function.

Similarly, the extrapolation from animal gut-derived VOCs to inter-organismal communication represents a significant conceptual leap unsupported by mechanistic evidence. While microbial communities produce gaseous metabolites, including SCFAs, sulfur-containing compounds, and phenolic derivatives [67,68], the assumption that these compounds influence “nearby individuals” through airborne transmission lacks rigorous experimental validation. Most studies demonstrating VOC effects utilize artificial experimental systems with unnaturally high concentrations and controlled atmospheres that bear little resemblance to natural conditions, where atmospheric dilution, chemical degradation, and background interference would severely limit signal transmission.

Furthermore, the claimed universality of SCFA signaling across kingdoms ignores fundamental physiological differences in receptor systems and metabolic pathways. While butyrate modulates histone deacetylase activity in mammalian cells, plants lack the specific G protein-coupled receptors (GPCRs) that mediate SCFA signaling in animals. Instead, plant responses to bacterial SCFAs appear to involve non-specific effects on cell membrane integrity and osmotic balance, mechanisms that are neither specific nor evolutionarily conserved [71]. This mechanistic divergence suggests that apparent cross-kingdom effects may represent convergent phenotypic outcomes rather than homologous signaling pathways.

The host-specificity of microbial metabolite responses is further exemplified by species-specific differences in cytochrome P450 enzymes, which determine metabolite bioavailability and activity. For instance, trimethylamine N-oxide (TMAO) production from gut bacterial trimethylamine varies 100-fold between humans and rodents due to differences in hepatic flavin monooxygenase expression, rendering animal model studies questionably relevant to human physiology [72]. This host-specific metabolic processing challenges any universal framework for microbial metabolite signaling.

Context-dependent outcomes represent another critical limitation in generalizing metabolite signaling functions. The immunomodulatory effects of bacterial metabolites like indole derivatives are highly dependent on the host immune status, age, and concurrent inflammatory conditions. In healthy individuals, these compounds may promote regulatory T cell responses, while in immunocompromised hosts, the same metabolites can exacerbate inflammatory cascades [73]. This context dependency suggests that microbial metabolites function more as modulatory factors within existing host regulatory networks than as universal signaling molecules with predictable cross-kingdom effects.

Furthermore, the temporal dynamics of metabolite signaling complicate universal interpretations. Bacterial production of signaling compounds follows circadian rhythms, nutrient availability cycles, and population density fluctuations, which may be completely misaligned with host physiological requirements. The half-life of most microbial VOCs ranges from minutes to hours, creating temporal windows where signaling is either absent or saturated. These conditions are incompatible with reliable inter-organismal communication [74].

Therefore, while microbial metabolites undoubtedly play important roles in host–microbiome interactions, their characterization as “universal signaling molecules” appears to be an oversimplification that conflates superficial similarities with mechanistic conservation. A more nuanced interpretation would recognize these compounds as context-dependent modulatory factors whose effects are constrained by host-specific physiology, environmental conditions, and evolutionary contingency.

## 6. How the Environment Influences Holobiont Colonization

The rhizosphere, characterized by its nutrient richness and microbial diversity, serves as a favorable niche for microbial proliferation. However, successful plant establishment depends on seed dispersal to unoccupied soils, minimizing competition. At germination, the seed-associated microbiome (spermosphere) is relatively limited. In experimental settings, germination can be studied using virus-free and microbe-free plantlets derived from surface-sterilized seeds or meristem-derived tissue cultures grown under sterile conditions. These axenic cultures are often used in plant biotechnology and plant–microbiome interaction studies to ensure the absence of endogenous viruses, bacteria, or fungi, enabling controlled inoculation and colonization experiments. These virus-free systems can be established using shoot tip cultures treated with antiviral agents or thermotherapy, followed by in vitro propagation on sterile media [75,76]. Once these virus-free plantlets are transferred to non-sterile environments, root exudates and environmental conditions shape the assembly of pioneer microbial communities in the rhizosphere, allowing the stepwise study of microbial recruitment and holobiont formation. Root colonization is initially shaped by stochastic processes, but ultimately leads to a rhizosphere community resembling that of conspecific plants, suggesting non-random assembly influenced by both environmental and seed-derived factors.

Seed germination releases specific exudates—such as sugars, amino acids, organic acids, and phenolic compounds—that selectively stimulate or inhibit microbial taxa. For instance, copiotrophs like *Pseudomonas* and *Enterobacter* are favored by nutrient-rich exudates, while secondary metabolites may suppress pathogens or favor beneficial colonizers [77]. Enzymes with antifungal properties, like chitinases, also indicate early active defense [78]. Some exudates, like malic acid, promote beneficial biofilm formation by microbes such as some species of the operational group of *Bacillus amyloliquefaciens* (OG*Ba*), which further modulates rhizosphere succession via secondary metabolite production [32,79]. In our laboratory, we observed that adding *B. velezensis* A6 to different plants protects those plants from drought by producing exopolysaccharides that improve soil moisture retention, activating stress-responsive genes, and inducing stomatal closure through ABA-mediated pathways to reduce water loss. In addition, the A6 strain produces a repertoire of antibiotics that control some pathogens. Importantly, these antibiotics promote the proliferation of other *Bacillus* strains with additional capabilities to protect these plants from drought, pathogens, and other stressors, and they produce VOCs that interact with other plants [32].

Soil physicochemical parameters (e.g., pH, redox potential, moisture) modulate interactions between the plant, the microbiota, and the environment, shaping a unique holobiont. Microbial recruitment pathways include abiotic vectors (wind, water) and biotic routes such as “bacterial highways” formed by arbuscular mycorrhizal fungi [80,81,82].

In parallel, animal (especially human) microbiome colonization is not purely stochastic, but instead influenced by congenital, inherited, and intrauterine factors. Humans serve as a model to contrast microbiome development between vaginal and cesarean births. In C-sections, the initial microbial exposure is largely hospital-derived. Nevertheless, considerable interindividual microbiome variation persists due to prenatal factors.

Embryonic and fetal development establishes a biochemical environment—partially modulated by the maternal microbiome—that predisposes the neonate to colonization by specific microorganisms. Maternal microbiota influence immune system priming, which guides postnatal microbial assembly. Additionally, the placenta, once considered sterile, hosts its own microbiome, contributing to maternal and fetal health and potentially conditioning neonatal microbial colonization [83,84].

## 7. How the Microbiome Comes to Dominate the Holobiont

In mammals, microbiome colonization occurs primarily via exposure to the maternal vaginal microbiota during birth, skin-to-skin contact, and lactation. Social behaviors, including physical contact, sharing surfaces (fomites), and sexual activity, further promote microbial exchange in social species.

Microorganisms residing in the gastrointestinal tract can modulate host feeding behavior through evolutionarily adaptive strategies that enhance their own fitness, potentially at the host’s expense. Ribeiro et al. [33] proposed two primary microbial strategies: inducing cravings for specific foods and generating dysphoria to compel nutrient intake. The mechanisms involved include the modulation of taste receptors, vagus nerve signaling, toxin production, and the synthesis of host-like neurochemicals and hormones across the lifespan. Notably, reduced microbial diversity is linked to increased obesity risk and maladaptive dietary habits [33]. This association may arise because a depauperate microbiome compromises metabolic flexibility, disrupting host energy homeostasis and satiety signaling. For instance, diminished SCFA-producing taxa (e.g., *Alistipes*, *Bifidobacterium*) impair gut barrier integrity and promote systemic inflammation, while obesogenic microbes may hijack neuroendocrine pathways to alter dietary preferences (e.g., cravings for high-fat foods). Such dysbiosis creates a feedback loop where diet further erodes microbial diversity, exacerbating metabolic dysfunction [85]. Ribeiro et al. also suggest that manipulating the microbiome via dietary interventions, prebiotics, probiotics, or fecal microbiota transplants may offer therapeutic avenues for managing obesity and related metabolic disorders.

Symbiosis between hosts and their microbiomes is viewed as a natural outcome of evolutionary processes, enabling rapid holobiont adaptation to environmental shifts. Microbiota–host interactions extend beyond individual adaptation, mediating communication and cooperation between holobionts. These inter-holobiont relationships are not confined to taxonomic boundaries, but instead form integrated ecosystems, where metabolic complementarity—akin to microbial cross-feeding—enhances the fitness of both the individual holobionts and the broader ecological community (Figure 3). These shared principles highlight how diverse holobiont systems converge on similar strategies of resilience, communication, and co-adaptation. To integrate these concepts, Table 3 provides a comparative overview of plant, animal, and human holobionts, summarizing their key microbial partners, core mechanisms, functional outcomes, and representative applications.

## 8. Conclusions

The holobiont paradigm has revolutionized our understanding of biological systems, revealing that stress adaptation, metabolism, and even behavior emerge from dynamic host–microbiome interactions. As demonstrated, microbial consortia drive ecosystem resilience, from methane-oxidizing bacteria enhancing plant drought tolerance to gut microbiota modulating neuroendocrine pathways in animals. However, critical gaps remain.

Current research prioritizes taxonomic profiling, but functional redundancy across microbiomes suggests that metabolic networks—not specific taxa—may dictate holobiont fitness. Future studies should leverage metatranscriptomics and metabolomics to dissect *how* microbiomes achieve plasticity, particularly in rapidly changing environments (e.g., climate-induced aridification).

Marine holobionts provide immediate opportunities for climate adaptation strategies. Coral probiotic therapies using thermotolerant *Symbiodinium* strains have demonstrated 2–4 °C increases in bleaching thresholds in field trials with *Acropora* species, while deep-sea mining operations could potentially harness extremophile partnerships for bioremediation [86,87]. Arctic systems face particular urgency, as permafrost thaw releases methane that could be mitigated through engineered partnerships between tundra plants and methanotrophic bacteria, building on natural associations already observed in *Eriophorum* species that reduce local methane emissions by up to 60% [88,89]. These examples demonstrate that holobiont engineering applications are not speculative but represent scalable interventions with demonstrated efficacy in natural systems.

The gut–brain axis exemplifies how microbial metabolites influence host physiology. Could future therapies combine probiotics with *personalized* microbiome modulators (e.g., phage cocktails or CRISPR-edited symbionts) to treat obesity or neurodegenerative diseases? Ribeiro et al.’s [33] study underscores this possibility, though host–microbiome coevolutionary conflicts (e.g., microbial “manipulation” of dietary cravings) complicate interventions.

If individual holobionts cooperate via volatile signals (e.g., plant–plant communication via microbial VOCs), we could speculate that entire ecosystems might function as “meta-holobionts” (Figure 4). This speculative framework could redefine conservation strategies, for instance, by prioritizing microbial dispersal corridors alongside species habitat protection.

The holobiont lens challenges reductionist biology, urging the integration of microbial ecology into medicine, agriculture, and climate science. As we advance, interdisciplinary collaboration spanning synthetic biology, systems ecology, and AI-driven modeling will be essential to harness holobiont dynamics for a sustainable future. However, we must remain vigilant, as manipulating these relationships poses risks with unintended consequences, demanding rigorous safeguards alongside innovation.

Microbiome engineering represents a transformative frontier with unprecedented potential to address global challenges in health, agriculture, and environmental restoration, while simultaneously presenting complex risks that demand careful consideration. The therapeutic applications show remarkable promise. Engineered *Lactobacillus* strains can deliver targeted pharmaceuticals directly to disease sites, reducing systemic side effects, while synthetic microbial consortia have shown efficacy in treating inflammatory bowel disease and metabolic disorders through precision metabolite production [90,91]. Agricultural applications offer equally compelling benefits, with drought-protective bacterial inoculants increasing crop yields by 20–40% under water stress conditions and engineered rhizosphere communities reducing fertilizer requirements by up to 30% through enhanced nutrient cycling [32,92].

Environmental restoration applications show particular promise, as methanotrophic bacterial partnerships can reduce greenhouse gas emissions while enhancing plant establishment in degraded soils, offering scalable climate mitigation strategies [31]. However, these benefits must be weighed against significant challenges, including horizontal gene transfer risks, where engineered genetic circuits can disseminate to native populations through conjugative plasmids within 48 h, potentially creating unintended ecological consequences [93,94]. Ecological trade-offs represent another concern, as beneficial modifications may disrupt established community networks. For example, the drought-protective *Bacillus* inoculants, while enhancing crop resilience, can simultaneously reduce soil mycorrhizal diversity and alter nitrogen cycling dynamics [95].

Ethical frameworks must evolve to address environmental releases where engineered microorganisms affect global ecosystem services, particularly regarding informed consent when microbial interventions can possibly horizontally transmit to communities without permission, as well as the potential displacement of Indigenous knowledge systems through technological dependence [96,97]. Realizing the transformative potential of microbiome engineering requires advancing containment-focused design principles, evolution-proof engineering approaches, and participatory governance frameworks that maximize benefits while ensuring reversibility, community autonomy, and long-term ecological stability. The holobiont paradigm necessitates a fundamental reconceptualization of environmental policy frameworks that have traditionally focused on individual species and discrete ecosystem components.

Climate resilience strategies must shift from protecting isolated organisms toward preserving and restoring holobiont networks that maintain ecosystem functionality under changing conditions. This requires integrating microbial diversity metrics into biodiversity indices, e.g., recognizing that a forest’s carbon sequestration capacity depends as much on the soil microbial consortia as on its plant species’ composition and interactions.

Agricultural policies must evolve beyond yield optimization to encompass soil microbiome health indicators, establishing payment schemes for ecosystem services that reward farmers for maintaining beneficial microbial diversity rather than merely for crop productivity.

Marine protected area design should prioritize microbial connectivity corridors that enable holobiont adaptation across temperature and pH gradients, while climate adaptation strategies must account for the fact that species range shifts are ultimately constrained by the dispersal and establishment capacity of their essential microbial partners.

Urban planning frameworks require the integration of “microbial infrastructure” considerations—green spaces designed not just for human recreation but also as reservoirs and corridors for beneficial microbiomes that support both ecosystem services and human health.

International climate agreements must recognize that methane mitigation strategies could leverage natural methanotrophic partnerships rather than relying solely on technological solutions, while biodiversity targets should include quantitative goals for maintaining functional microbial diversity alongside traditional species metrics.

Perhaps most critically, environmental impact assessments must incorporate holobiont-scale analyses that evaluate how proposed developments will affect not just visible species but also the microbial networks that underpin ecosystem resilience, recognizing that disrupting these invisible partnerships may have cascading consequences that manifest only across ecological timescales. This holobiont-informed policy framework would represent a paradigm shift from managing individual components toward stewarding the microbial foundations of planetary health, acknowledging that, in an era of rapid environmental change, our survival depends on maintaining the ancient partnerships that have enabled life’s persistence through previous global transitions.

## Figures and Tables

**Figure 1 biology-14-01203-f001:**
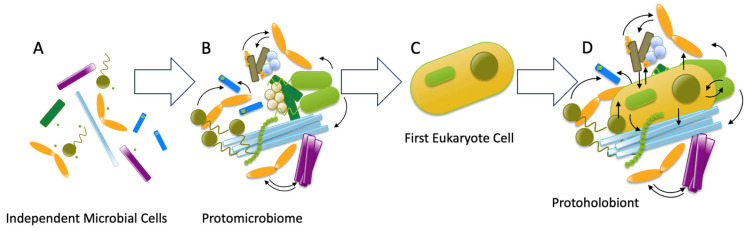
Proposed evolution of holobionts. (**A**) First microorganism interacting with environment. (**B**) Formation of *protomicrobiomes* by establishing metabolic and genetic interaction (represented with narrow arrows) between the different microorganisms. (**C**) Appearance of the first eukaryote by endosymbiosis. (**D**) Establishment of protoholobiont by metabolic and genetic interactions between the different microorganisms and the newly generated eukaryote. Image was created with PowerPoint 16.78.3.

**Figure 2 biology-14-01203-f002:**
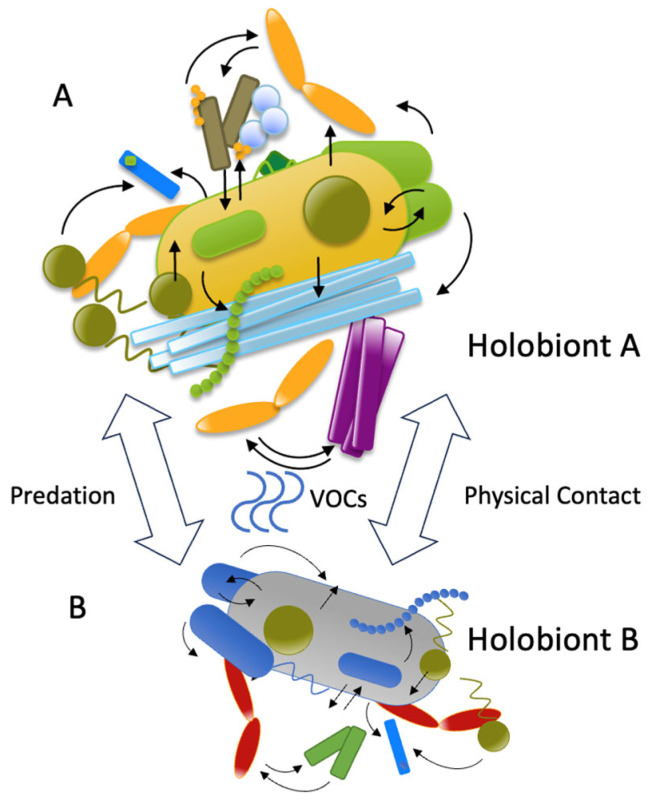
Cross-species microbial transfer with integrated inter-holobiont signaling. (**A**) Holobiont A can influence the microbial composition of (**B**) Holobiont B through the predation of A, the production of metabolites such as VOCs, or physical contact. Image was created with PowerPoint.

**Figure 3 biology-14-01203-f003:**
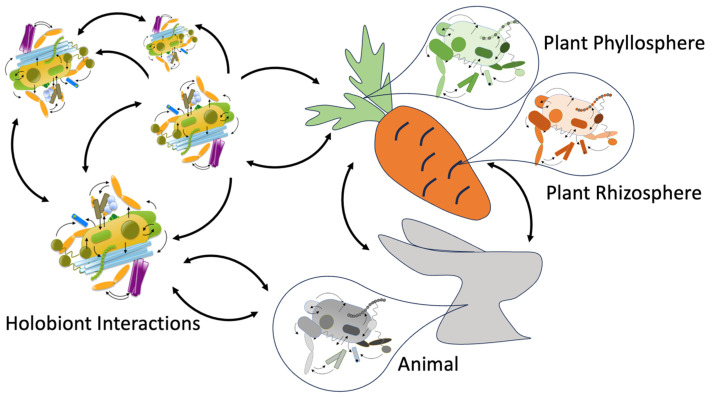
Inter-holobiont relationships in ecosystems. Inter-species and cross-kingdom interactions of holobionts. Image was created with PowerPoint.

**Figure 4 biology-14-01203-f004:**
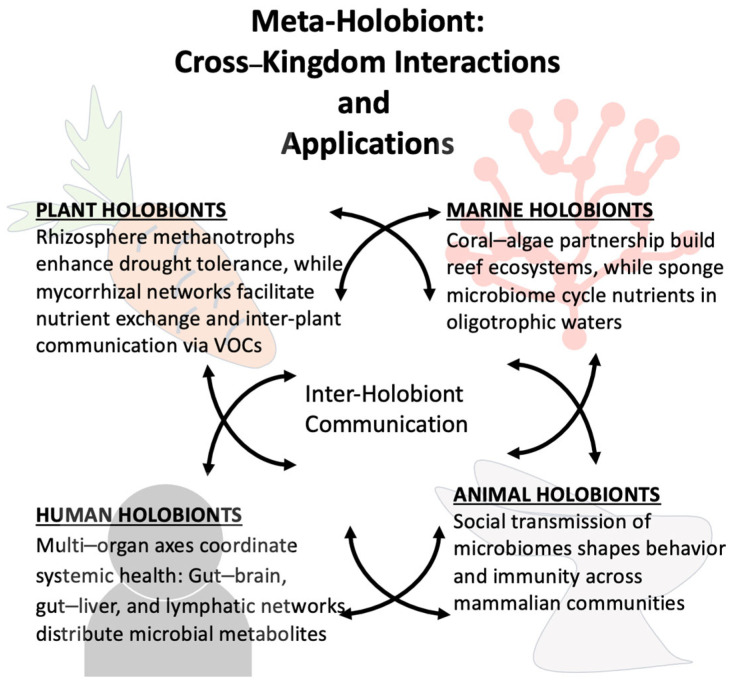
Meta-holobiont as a cross-kingdom network of holobiont interactions. Conceptual illustration on how individual holobionts interact across taxonomic boundaries to form integrated ecological networks.

**Table 1 biology-14-01203-t001:** Holobiont adaptations to abiotic and biotic stresses.

Stress Type	Host System	Microbiome Role	Mechanism	Outcome	Reference in Text
Drought	Plants	Methanotrophs (e.g., *Methylobacterium*)	Oxidize CH_4_ → CO_2_ + H_2_O; methanol production reshapes rhizosphere	Enhanced water retention, stress-responsive genes	Delherbe et al. (2025) [31]
Pathogen Defense	Plants	*Bacillus velezensis* A6	Antibiotic production, VOC signaling, induction of systemic resistance	Reduced pathogen load, improved survival	Barros-Rodríguez et al. (2024) [32]
Nutrient Limitation	Plants	Rhizosphere microbiota (e.g., *Pseudomonas*)	Phosphate solubilization, siderophore production	Improved nutrient uptake	Manzanera et al. (2015) [27]
Obesity	Humans	Gut microbiota (e.g., *Alistipes*, *Bifidobacterium*)	SCFA production, gut-barrier integrity, modulation of satiety signals	Metabolic homeostasis	Ribeiro et al. (2025) [33]
Neurodegeneration	Humans	Gut–brain axis microbes (e.g., *Lactobacillus*)	Neurotransmitter synthesis (e.g., GABA, s serotonin), immune modulation	Mood regulation, reduced neuroinflammation	Solari et al. (2021) [34]

**Table 2 biology-14-01203-t002:** Major types of host–microbiome interactions, representative examples, and their ecological and physiological outcomes.

Type of Interaction	Representative Examples	Mechanisms	Benefits/Costs for Host	Benefits/Costs for Microbe
Mutualism (obligate)	Mycorrhizal fungi–plant; Rhizobia–legumes	Nutrient exchange (P solubilization, N_2_ fixation)	Enhanced nutrient uptake, growth promotion, stress resilience	Access to carbon, stable habitat
Mutualism (facultative)	Plant growth-promoting rhizobacteria (*Azospirillum, Bacillus*)	Hormone production (auxins, cytokinins), ACC deaminase activity	Improved root growth, tolerance to abiotic stress (drought, extreme temperatures)	Rhizosphere colonization, metabolic byproducts used
Commesalism	Endophytic bacteria (*Enterobacter* spp.)	Colonization without major host effect	Neutral; sometimes, indirect growth promotion	Shelter; nutrient leakage
Context-dependent symbiosis	*Pseudomonas* spp. (beneficial vs. opportunistic)	Gene regulation depending on host stress	Can shift from growth promotion to pathogenicity	Survival flexibility under changing host conditions
Antagonism (pathogenicity)	*Phytophtora infestans* (late blight in potatoes)	Tissue invasion, effector secretion	Tissue damage, yield loss	Nutrient extraction, proliferation
Cooperation (microbe–microbe)	*Pseudomonas–Trichoderma* synergism	Cross-feeding, quorum sensing, co-biofilm formation	Enhanced pathogen defense, stress resilience	Expanded niche, survival advantage
Environmental engineering	Methanotrophs in arid soils with plants	Oxidation of CH_4_ → CO_2_ + H_2_O	Improved water and carbon availability	Stable ecological niche, energy source

**Table 3 biology-14-01203-t003:** Representative microbiome–lymph node associations and their physiological implications.

Anatomical Region	Microbial Reservoirs (Examples)	Associated Lymph Node	Representative Mechanisms	Potential Physiological Outcomes
Gut/Mesentery	Intestinal microbiota (*Bacteroides*, *Lactobacillus*, *Clostridia*)	Mesenteric lymph nodes	Microbial metabolite transport via lymph and portal vein; IgA production	Modulation of hepatic metabolism, immune tolerance, systemic inflammation control
Cervical region	Tonsillar crypts (*Fusobacterium*, *Prevotella*), nasopharyngeal mucosa (*Streptococcus pneumoniae*, *Neisseria*), salivary glands (*Streptococcus*, *Veillonella*)	Cervical lymph nodes	Sampling of oral/nasal antigens; induction of mucosal IgA	Local mucosal immunity, regulation of airway inflammation, systemic immune priming
Axillary region	Apocrine glands (*Corynebacterium*, *Staphylococcus epidermidis*), hair follicles (*Cutibacterium acnes*, *Malassezia*), sebaceous glands (*Staphylococcus hominidis*)	Axillary lymph nodes	Antigen drainage from skin; microbial metabolite processing	Regulation of cutaneous immunity, modulation of host odor cues, pathogen defense (e.g., *S. aureus*)
Inguinal region	Perineal and genital microbiota (*Lactobacillus*, *Gardnerella*)	Inguinal lymph nodes	Antigen sampling from mucocutaneous junctions	Maintenance of genital tract immune balance, protection against urogenital pathogens

## Data Availability

No new data were created or analyzed in this study. Data sharing is not applicable to this article.

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
