# Peer review of "From Microbial Consortia to Ecosystem Resilience: The Integrative Roles of Holobionts in Stress Biology"

_biology, 2025, doi:10.3390/biology14091203_

Round 1
Reviewer 1 Report
Comments and Suggestions for Authors
The review article summarized the current achievements about the integrative roles of the holobionts in stress biology. Generally, the MS provides interesting information for better understanding that holobionts from plants to animals, rely on microbial partnerships to thrive under environmental challenges. I believe that the MS may contributions to the related field and mechanism of influence of the holobionts. However, there are still several issues that need to be modified.
- Check and improve some sentences in the MS, for example, line 91-92, "In early Earth’s anoxic environment, dominated solely by microorganisms, a dynamic equilibrium favored metabolic adaptability." . Line 147, "Image created with PowerPoint ", it should "Image was created with PowerPoint. "
- Section 2, from line 172-219, it contains only one paragraph, suggest to divided this part into several paragraphs to make it clearly to read. Also, for line 267-293.
- Only two figures and one table in the MS. Suggest to add more figures and tables to make the article rich in content and easy to understand.
- For the two figures in the MS, the current content of the figures are relatively simple, and the information provided is limited. It is recommended to modify it and add more abundant contents.
- Line 30, secondly, add a comma.
- Line 39, give the full name for SCFAs, when it appears for the first time in the MS.
- Line 76, no period at the end of the sentence. Please check.
- Line 220, check the title for format inconsistency.
- Line 315, VOCs, not need to give the full name here.
- References: please correct many formaterrors. Also, the number of the references is not enough. It suggest to add more influential articles. The current 63 references seems not enough for covering this important topic.
- Table 1. please recheck the format of the table. Also, the informationin this table is limited. Please revise and expand this table.
The English could be improved to more clearly express the research.
Author Response
|
Comments 1: [Check and improve some sentences in the MS, for example, line 91-92, "In early Earth’s anoxic environment, dominated solely by microorganisms, a dynamic equilibrium favored metabolic adaptability."]
|
|
|
Comments 2: Line 147, "Image created with PowerPoint ", it should "Image was created with PowerPoint.] |
|
Response 2: Agree. I have, accordingly, modified the sentences as legends to the figures 1 and 2 in lines 206 and 445.
Comments 3: [Section 2, from line 172-219, it contains only one paragraph, suggest to divided this part into several paragraphs to make it clearly to read. Also, for line 267-293.]
Response 3: Thank you again for pointing this out. Based on your comment, I have divided the Section 2 into 6 different paragraphs so it easier to read now. Also, I have divided the other paragraph (formerly lines 267 to 293) into 5 new paragraphs that I think that makes the text easier to read as well.
Comment 4 [Only two figures and one table in the MS. Suggest to add more figures and tables to make the article rich in content and easy to understand.] Response 4: Indeed, you are right that it was necessary to add two additional figures (Figure 3 and Graphical Abstract). These extra figures enrich the article in content and makes the article easier to understand. Also two additional Tables (Table 1 and Table 2) have been included.
Comment 5 [For the two figures in the MS, the current content of the figures are relatively simple, and the information provided is limited. It is recommended to modify it and add more abundant contents.]
Response 5: Thank you for your suggestion. I think doubling the number of images improves the content, making it more complex and expanding the information. Thanks for the suggestion.
Comment 6 [Line 30, secondly, add a comma]
Response 6: Thank you for pointing this out. I have hired the MDPI editing service to review the whole article to avoid mistakes such as this one.
Comment 7 [Line 39, give the full name for SCFAs, when it appears for the first time in the MS.]
Response 7: Previous response also applies here.
Comment 8 [Line 76, no period at the end of the sentence. Please check.]
Again, previous response also applies here. Thank you again.
Comment 9 [Line 220, check the title for format inconsistency.]
Response 9: Thank you very much. The title has been modified to avoid formatting inconsistencies.
Comment 10 [Line 315, VOCs, not need to give the full name here.]
Response 10: Thank you very much, the error has now been corrected.
Comment 11 [References: please correct many format errors. Also, the number of the references is not enough. I suggest to add more influential articles. The current 63 reference seems not enough for covering this important topic.]
Response 11: I appreciate the reviewer’s comment and have checked for format errors. The former number of 63 references have been increased.
Comment 12 [Table 1. please recheck the format of the table. Also, the information in this table is limited. Please revise and expand this table.]
Response 12: I thank the reviewer for raising this issue. As previously mentioned, and in order to deal with the expansion two additional tables (Table 1 and Table 2) have been included.
|
|
4. Response to Comments on the Quality of English Language |
|
Point 1: The English could be improved to more clearly express the research. |
|
Response 1: I appreciate your comment in this respect. The editing service has been hired, to improve the quality of English language. |

Reviewer 2 Report
Comments and Suggestions for Authors
This opinion presents a compelling synthesis that successfully reframes stress biology through the lens of holobiont theory. The integration of evolutionary, mechanistic, and ecological perspectives is intellectually stimulating and well aligned with the current interdisciplinary focus in microbiome research. However, the article would benefit from more concrete examples that bridge theory with experimental evidence, especially in underexplored systems, such as marine holobionts or extremophiles.
- The article effectively spans diverse biological domains (plants and animals), yet the transitions between sections occasionally appear abrupt. Clarifying section headings or using subheadings could improve readability and guide the reader through the evolutionary-to-ecological continuum. Additionally, explicitly distinguishing between opinion-based extrapolation and consensus-supported conclusions would enhance clarity.
- The framing of microbial metabolites (e.g., SCFAs and VOCs) as universal signaling molecules across kingdoms is novel and intriguing. However, the article might overstate the generalizability of these metabolites without addressing known limitations such as host-specific responses or context-dependent outcomes. A more critical discussion strengthens this argument.
- The discussion of microbiome engineering and probiotics is timely and relevant. However, the treatment was somewhat superficial in comparison to the rest of the article. Expanding on current challenges, such as horizontal gene transfer risks, ecological trade-offs, and ethical concerns, would provide a more nuanced approach to these biotechnological solutions.
- The article’s ambition to unite evolutionary biology, microbiology, and ecology within the holobiont framework is laudable. To further strengthen this interdisciplinary relevance, we briefly discuss how this perspective may influence environmental policies, climate resilience strategies, and biodiversity conservation efforts. A forward-looking paragraph in the conclusion can significantly enhance this impact.
- The English in MS needs to be greatly improved.
- Ensure that all abbreviations are defined at first use in both the abstract and the main text and used consistently throughout the manuscript to improve readability.
- The English in MS needs to be greatly improved.
Author Response
|
Comments 1: [This opinion presents a compelling synthesis that successfully reframes stress biology through the lens of holobiont theory. The integration of evolutionary, mechanistic, and ecological perspectives is intellectually stimulating and well aligned with the current interdisciplinary focus in microbiome research. However, the article would benefit from more concrete examples that bridge theory with experimental evidence, especially in underexplored systems, such as marine holobionts or extremophiles.]
|
|
|
Comments 2: [The article effectively spans diverse biological domains (plants and animals), yet the transitions between sections occasionally appear abrupt. Clarifying section headings or using subheadings could improve readability and guide the reader through the evolutionary-to-ecological continuum. Additionally, explicitly distinguishing between opinion-based extrapolation and consensus-supported conclusions would enhance clarity.] |
|
Response 2: Thank you for pointing this out. I have tried to avoid abrupt transitions and clarify sections.
Comments 3: [The framing of microbial metabolites (e.g., SCFAs and VOCs) as universal signaling molecules across kingdoms is novel and intriguing. However, the article might overstate the generalizability of these metabolites without addressing known limitations such as host-specific responses or context-dependent outcomes. A more critical discussion strengthens this argument.]
Response 3: Thank you again for pointing this out. Based on your comment, I have modified the discussion section, in specific Section 4 (lines 491 to 649) to avoid the generalization of SCFA and VOCs as universal signaling molecules. I hope you find this new section appropriate critical discussion.
Comment 4 [The discussion of microbiome engineering and probiotics is timely and relevant. However, the treatment was somewhat superficial in comparison to the rest of the article. Expanding on current challenges, such as horizontal gene transfer risks, ecological trade-offs, and ethical concerns, would provide a more nuanced approach to these biotechnological solutions.]
Response 4: Indeed, you are right that it was necessary to add additional discussion with reference to current challenges such as horizontal gene transfer risks, ecological trade-offs, and ethical concerns. Such text is now included (lines 819 to 850).
Comment 5 [The article’s ambition to unite evolutionary biology, microbiology, and ecology within the holobiont framework is laudable. To further strengthen this interdisciplinary relevance, we briefly discuss how this perspective may influence environmental policies, climate resilience strategies, and biodiversity conservation efforts. A forward-looking paragraph in the conclusion can significantly enhance this impact.]
Response 5: Thank you for your suggestion. I have included the last 32 lines of the Conclusion section to further strengthen the interdisciplinary relevance on the article in uniting all indicated fields.
Comment 6 [Ensure that all abbreviations are defined at first use in both the abstract and the main text and used consistently throughout the manuscript to improve readability.].
Response 6: Thank you very much, I have reviewed the article to avoid this type of errors.
|
|
4. Response to Comments on the Quality of English Language |
|
Point 1: The English in MS needs to be greatly improved. |
|
Response 1: I appreciate your comment in this respect. Nevertheless, an editing service has been hired, to improve the quality of English language. Please find the attached certificate. |

Reviewer 3 Report
Comments and Suggestions for Authors
The manuscript is an attempt to review the literature related to the role of holobionts in stress management by the living systems. The topic of the article is very broad and the authors tried to cover different aspects superficially. In depth discussion on the topic is missing in the article. Specifically,
- The scope of the article should be clearly stated in the abstract and then in the introduction.
- There are many recent reviews on different aspects of holobionts. Authors should argue about the need of this review. They should explicitly state how this review standout among the existing papers.
- An introduction section should be there to provide a broad perspective of the field and the article. Authors should state about the stress conditions and their impacts on living forms.
- There are many statements that are written without proper citations. References should be provided wherever important information has been given. It should be checked throughout the manuscript.
- L59: Correct this sentence
- L74: Provide reference. Also argue its importance
- Section 3 and 4 should be extended. The source of microbiome, their establishment and typical phyla, genera or species should be given. Authors should discuss how the microbiome and their host shape each other.
- L162-170: Personal communication should not be references in a review article. Any such information should be given from a reliable source and personal communication should be referred as a supportive statement.
- L198-199: Provide reference. What could be the possible reason?
- The section final remarks should be replaced with conclusion and future outlook. Authors should speculate their view/opinion on this field here.
- Generally, it is expected that the authors of a review article have out efforts in reviewing many research articles and the bibliography list would be much longer than 25 references.
- There should be some illustrations and tables to better communicate the message of the article. A detailed illustration/table presentation of the studies related to stresses and holobionts association should be summarized.
Comments on the Quality of English Language
Minor language editing is required.
Author Response
For research article
|
Response to Reviewer 1 Comments
|
||
|
1. Summary |
|
|
|
Thank you very much for taking the time to review this manuscript. Please find the detailed responses below and the corresponding corrections in track changes in the re-submitted files. I appreciate your comments on my manuscript and agree in modify as you suggest, as I think this makes a much better article. mistake.
|
||
|
2. Questions for General Evaluation |
Reviewer’s Evaluation |
Response and Revisions |
|
Does the introduction provide sufficient background and include all relevant references? |
Must be improved |
Thank you. The Introduction has been extended considerably. I understand that now it provides sufficient background and include all relevant references. |
|
Are all the cited references relevant to the research? |
Not applicable |
|
|
Is the research design appropriate? |
Not applicable |
|
|
Are the methods adequately described? |
Not applicable |
|
|
Are the results clearly presented? |
Not applicable |
|
|
Are the conclusions supported by the results? |
Must be improved |
Thank you. The conclusions have been extended as well. I understand that it is now to the referee´s satisfaction. |
|
3. Point-by-point response to Comments and Suggestions for Authors |
||
|
Comments 1: [The scope of the article should be clearly stated in the abstract and then in the introduction.]
|
||
|
Response 1: Thank you for pointing this out. I agree with this comment. Therefore, I have rewritten the abstract and included an introduction describing the scope of the article so the topic of the article is narrower now. With respect to the Abstract, I have added a clear three-part scope (evolution, mechanisms, ecology) to structure the review’s contributions. Also, I have explicitly named microbial metabolites as a unifying theme. Then I have sharpened the translational focus on microbiome engineering. With respect to the Introduction, I have positioned the work as ing gaps in prior literature (evolutionary, mechanistic, and in reference to ecological integration). I also have contrasted plant/animal systems to highlight cross-kingdom insights. Finally, I have emphasized the paradigm shift from host-centric to holobiont-centric adaptation.
You can find these modifications in page 1 and 2.
|
||
|
Comments 2: [There are many recent reviews on different aspects of holobionts. Authors should argue about the need of this review. They should explicitly state how this review standout among the existing papers.] |
||
|
Response 2: We appreciate the referee’s insightful observation and agree that the growing body of literature on holobionts necessitates a clear justification for this review. Our work stands out by integrating three underexplored dimensions that collectively advance holobiont theory beyond existing reviews:
Comments 3: [An introduction section should be there to provide a broad perspective of the field and the article. Authors should state about the stress conditions and their impacts on living forms.]
Response 3: Thank you again for pointing this out. I have included an Introduction section to provide a broad perspective on both, on the field and on the article.
Comment 4 [There are many statements that are written without proper citations. References should be provided wherever important information has been given. It should be checked throughout the manuscript.] Response 4: Indeed, you are right that it was necessary to check the references. References have being checked throughout the manuscript.
Comment 5 [L59: Correct this sentence]
Response 5: Thank you for pointing to the correction of the sentence. The sentence has been corrected to “These protoholobionts represent an evolutionary transition toward complex holobionts—multicellular eukaryotes forming obligate partnerships with phylogenetically diverse microbiota (bacteria, archaea, fungi, viruses, and protists)—where host-microbiome metabolic complementarity established a functionally integrated metaorganism with emergent ecological capabilities.”, as you can find in Line 98.
Comment 6 [L74: Provide reference. Also argue its importance]
Response 6: I really appreciate your point. The sentence to previous L74 has been rewritten that helps me to explicit functional roles, clarifying how prokaryotic genes contribute (e.g., metabolism, immunity) rather than just stating they are “significant.”. Also, by giving examples, I add concrete biological processes (vitamin synthesis, detoxification) to anchor the claim. And finally, I can include an evolutionary context as the newly included text ties the gene repertoire to adaptive flexibility, aligning with the paper’s them of microbiome-driven resilience. Please find the new text in last paragraph of the third page (lines 115-121). In addition, I have aimed fo specificity, by giving names of concrete metabolites (SCFAs, GABA, indole) and their roles. I have dealed with mechanistic clarity by explaining how signals reach the CNS (e.g., serotonin synthesis, quorum sensing), giving in all cases specific references. And finally, I have aligned with the themes that reinforces the gut-brain axis (Section 3) and the stress adaptation (Abstract).
Comment 7 [Section 3 and 4 should be extended. The source of microbiome, their establishment and typical phyla, genera or species should be given. Authors should discuss how the microbiome and their host shape each other.]
Response 7: I understand your concerns and I have extended both Sections (3 and 4) including the source of the microbiome, their establishment and typical phyla, genera or species.
Comment 8 [L162-170: Personal communication should not be references in a review article. Any such information should be given from a reliable source and personal communication should be referred as a supportive statement.]
Response 8: I thank the reviewer for this insightful question. In consequence, this text has been reformulated and reference to Personal communication has been removed. See the new text in first paragraph of page 8 (current L340-348).
Comment 9 [L198-199: Provide reference. What could be the possible reason?]
Response 9: Upon request of the reviewer a paragraph has been included to describe the possible reason that links dysbiosis to dietary choices, aligning with the holobiont concept in the manuscript. I have included two references (Ribeiro et al. (2025) and Chanda and De, (2025)) which directly discusses microbiome-driven appetite modulation. The paragraph also refers to the possible mechanism of action, and highlights loss of beneficial taxa (SCFA producers) and pathogenic microbial strategies (e.g., cravings). This integrates seamlessly with the paper’s themes of microbiome-host coevolution and stress adaptation.
Comment 10 [The section final remarks should be replaced with conclusion and future outlook. Authors should speculate their view/opinion on this field here.]
Response 10: I thank again the reviewer for this useful suggestion. I have replaced the Final Remarks section with a Conclusion and Future Outlook section. In this new section I have speculate with my opinion on this field, replacing descriptive observations with forward-looking, opinion-driven speculation. Also, I have emphasized applications (e.g., engineered symbiosis) and risks (e.g., ecological disruptions). In addition, I have anchored ideas in cited work (e.g., Ribeiro et al.) while introducing novel hypotheses. And finally, I have aligned with the paper’s themes of adaptation, cooperation, and microbiome-host coevolution to your satisfaction.
Comment 11 [Generally, it is expected that the authors of a review article have out efforts in reviewing many research articles and the bibliography list would be much longer than 25 references.]
Response 11: I appreciate the reviewer’s comment and have reviewed many research articles. This has resulted in a longer Review article where consequently the bibliography list has been expanded from 25 references to the current 63.
Comment 12 [There should be some illustrations and tables to better communicate the message of the article. A detailed illustration/table presentation of the studies related to stresses and holobionts association should be summarized.]
Response 12: I thank the reviewer for raising this comment. Two figures have been included to depict early formation of holobionts and the holobionts associations. In addition, a table has been included summarizing key examples from the manuscript with specific references.
|
||
|
4. Response to Comments on the Quality of English Language |
||
|
Point 1: The English could be improved to more clearly express the research. |
||
|
Response 1: I appreciate your comment in this respect. An effort has been made to improve the English of the manuscript to comply with the minor language editing is required, as mentioned by this reviewer.
|
||
|
5. Additional clarifications |
||
|
N/A |
||

Reviewer 4 Report
Comments and Suggestions for Authors
The author has presented an interesting paper in the "Opinion" section. I would recommend transforming it into a Review, since many of the author's statements are overly categorical and unsubstantiated. Also, mainly private facts are described that are not typical for all living organisms and their systems. We know of many cases where the holobiont concept does not work, for example, when obtaining virus-free plants from cell cultures.
Comments on the text of the manuscript:
The manuscript contains two sets of keywords.
Line 32: the emergence of life on Earth is a process described mainly at the level of hypotheses. We do not know whether it arose quickly or slowly, whether it arose in the form of individual organisms or in the form of ecosystems. The author does not need to be so categorical.
Lines 51-52: what does the author mean by metabolically integrated microbial consortia? Can you give a specific example of them?
Line 55: it is necessary to give an example of free-living functionally specialized microbiomes.
Lines 71-74: what contribution of prokaryotic genes are we talking about? Contribution to what?
Line 77: can you give several examples of rapid metabolic adjustments and indicate which specific chemicals transmit the signal to the central nervous system.
Line 78: can you give examples of genetic modifications at the eukaryotic level that are being discussed? How does the microbiome react to infectious diseases of animals and humans?
Line 83: are absolutely all rhizosphere microorganisms mutualists? Are there commensals there, and most importantly, cooperative relationships?
Lines 100-103: i.e. bacterial communities can exist separately from plants, perhaps not evolutionary, but rapid formation of a holobiont? How does the microbiome know the "success" of its own metabolic pathways for the evolution of eukaryotes?
Line 115: how many such axes does an animal have? Can you provide links to describe each of them?
Lines 124-125: What places of microbial concentration does the author talk about when he mentions the cervical and axillary regions?
Lines 133-134: Do microbiomes inside a person also affect his body by producing volatile organic compounds? Provide examples from the literature.
Lines 149-150: How does germination from virus-free cell cultures occur?
The article is interesting and thought-provoking. However, not all of the processes described are widespread and have evolutionary significance. In addition, the author neglects the laws of logic and some biological facts.
Author Response
For research article
|
Response to Reviewer 2 Comments
|
||
|
1. Summary |
|
|
|
Thank you very much for taking the time to review this manuscript. Please find the detailed responses below and the corresponding revisions/corrections highlighted/in track changes in the re-submitted files. We appreciate that you consider our manuscript as an interesting paper and agree in removing one of the two sets of keywords introduced by mistake.
|
||
|
2. Questions for General Evaluation |
Reviewer’s Evaluation |
Response and Revisions |
|
Does the introduction provide sufficient background and include all relevant references? |
Yes |
Thank you. [Please give your response if necessary. Or you can also give your corresponding response in the point-by-point response letter. The same as below] |
|
Are all the cited references relevant to the research? |
Yes/Can be improved/Must be improved/Not applicable |
|
|
Is the research design appropriate? |
Yes/Can be improved/Must be improved/Not applicable |
|
|
Are the methods adequately described? |
Yes/Can be improved/Must be improved/Not applicable |
|
|
Are the results clearly presented? |
Yes/Can be improved/Must be improved/Not applicable |
|
|
Are the conclusions supported by the results? |
Yes/Can be improved/Must be improved/Not applicable |
|
|
3. Point-by-point response to Comments and Suggestions for Authors |
||
|
Comments 1: [The manuscript contains two sets of keywords.]
|
||
|
Response 1: Thank you for pointing this out. I agree with this comment. Therefore, I have removed one of the two sets of keywords introduced by mistake.
|
||
|
Comments 2: [Line 32: the emergence of life on Earth is a process described mainly at the level of hypotheses. We do not know whether it arose quickly or slowly, whether it arose in the form of individual organisms or in the form of ecosystems. The author does not need to be so categorical.] |
||
|
Response 2: Agree. I have, accordingly, modified that paragraph to emphasize this point within a hypothese context. As you can see now text from line 31 to 42 has being modified in the last paragraph of page 1 with softened assertations, highlighting hypotheses while retained the citations.
Comments 3: [Lines 51-52: what does the author mean by metabolically integrated microbial consortia? Can you give a specific example of them?]
Response 3: Thank you again for pointing this out. Based on your comment, I have changed the text to state that In Lines 51-52, I refer to "metabolically integrated microbial consortia" as communities of microorganisms that interact synergistically through the exchange of metabolites and genetic material, creating a functionally interdependent network. These consortia are characterized by cross-feeding relationships, where the metabolic byproducts of one organism serve as substrates for another, enabling collective survival and efficiency in nutrient cycling or energy acquisition. I hope that I have improved clarity by explicitly defining the term metabolically integrated with a concrete example, while maintaining the original narrative and embedding the example naturally, to your satisfaction.
Comment 4 [Line 55: it is necessary to give an example of free-living functionally specialized microbiomes.] Response 4: Indeed, you are right that it was necessary to give an example of free-living functionally specialized microbiomes, that we believe it is now included in the text within the second paragraph of page 2, current lines 56-60.
Comment 5 [Lines 71-74: what contribution of prokaryotic genes are we talking about? Contribution to what?]
Response 5: Thank you for pointing to the comment that helps me to explicit functional roles, clarifying how prokaryotic genes contribute (e.g., metabolism, immunity) rather than just stating they are “significant.”. Also, by giving examples, I add concrete biological processes (vitamin synthesis, detoxification) to anchor the claim. And finally, I can include an evolutionary context as the newly included text ties the gene repertoire to adaptive flexibility, aligning with the paper’s them of microbiome-driven resilience. Please find the new text in last paragraph of the second page (lines 80 - 86). In addition, I have aimed fo specificity, by giving names of concrete metabolites (SCFAs, GABA, indole) and their roles. I have dealed with mechanistic clarity by explaining how signals reach the CNS (e.g., serotonin synthesis, quorum sensing). And finally, I have aligned with the themes that reinforces the gut-brain axis (Section 3) and the stress adaptation (Abstract).
Comment 6 [Line 78: can you give examples of genetic modifications at the eukaryotic level that are being discussed? How does the microbiome react to infectious diseases of animals and humans?]
Response 6: I really appreciate your point and examples of genetic modifications are being discussed now. For the eukaryotic modifications, examples on epigenetic changes (DNA methylation), on somatic recombination (immune genes) and on allele selection (lactase persistence), have been introduced in the text (see Line 100 – 112). These modifications describe the microbiome responses to infection, in terms of horizontal gene transfer, phyla-level shift, and production of antimicrobial, giving a sense of temporal framework, to highlight the contrasts of immediate microbial vs. gradual eukaryotic responses.
Comment 7 [Line 83: are absolutely all rhizosphere microorganisms mutualists? Are there commensals there, and most importantly, cooperative relationships?]
Response 7: I understand your concerns and have modified the text include other relationship apart from mutualistic such as commensal and cooperative relationships, that can be found in the last paragraph of page 3. In this way, explicitly acknowledges mutualists, commensals, and conditional pathogen, providing specific examples (Pseudomonas, Enterobacter). Also, address the cooperation mechanisms in a contexto f cross-feeding (metabolic interdependence), biofilm coordination (quorum-sensing) and points to the collective pathogen suppression. Therefore the new text highlights strain-specific behavioral shifts and maintains focus on net benefits to host.
Comment 8 [Lines 100-103: i.e. bacterial communities can exist separately from plants, perhaps not evolutionary, but rapid formation of a holobiont? How does the microbiome know the "success" of its own metabolic pathways for the evolution of eukaryotes?]
Response 8: Trying to respond to the comment 8, I have included information regarding the rapid holobiont formation, that explicitly states bacterial independence from plants; that details how transient consortia form via chemoattraction (malate/luteolin). Also, an evolutionary mechanism that considers the fitness feedback loops as the selection driver is discussed. As well as specifies molecular determinants (ACC deaminase for stress tolerance)
Comment 9 [Line 115: how many such axes does an animal have? Can you provide links to describe each of them?]
Response 9: Upon request of the reviewer a paragraph has been included to describe other axes a part from the gut-brain axes. Names of exact metabolites (TMAO, SCFAs) and pathways (portal circulation, immune trafficking) are mentioned. This paragraph involves a logic flow to integrate the orders axes by mechanistic clarity, connecting liver with lung, with skin and finally with heart.
Comment 10 [Lines 124-125: What places of microbial concentration does the author talk about when he mentions the cervical and axillary regions?]
Response 10: I thank the reviewer for this insightful question. To clarify, the cervical and axillary regions refer to anatomical areas that are adjacent to mucosa- and skin-associated microbiomes. These regions contain dense lymphatic structures (e.g., cervical and axillary lymph nodes) that drain microbe-rich areas such as the oral cavity, upper respiratory tract, and moist skin folds (e.g., armpits). Recent studies have shown that commensal microbes, or their metabolites, can reach these lymph nodes, where they influence immune function and systemic physiology. We have revised the manuscript accordingly to reflect this clarification (now included in Section 3).
Comment 11 [Lines 133-134: Do microbiomes inside a person also affect his body by producing volatile organic compounds? Provide examples from the literature.]
Response 11: I appreciate the reviewer’s comment and have added a clarifying paragraph to address this important point. Indeed, endogenous human microbiomes—particularly in the gut and oral cavities—produce a wide range of volatile organic compounds (VOCs) that modulate host physiology and can serve as biomarkers for health status. Several examples from the literature have been included to support this statement.
Comment 12 [Lines 149-150: How does germination from virus-free cell cultures occur?]
Response 12: I thank the reviewer for raising this question. Germination from virus-free cell cultures refers to the process by which axenic (microbe-free) plantlets or seedlings are obtained from surface-sterilized or in vitro-propagated tissues, often used in plant biotechnology and microbiome research. These virus-free (and often pathogen-free) systems enable the study of initial microbial colonization dynamics in a controlled environment. A clarifying paragraph and appropriate references have now been added to Section 5.
|
||
|
4. Response to Comments on the Quality of English Language |
||
|
Point 1: The English is fine and does not require any improvement. |
||
|
Response 1: I appreciate your comment in this respect. |
||
|
5. Additional clarifications |
||
|
I appreciate as well that the review considers the article as interesting and thought-provoking, as this was the original intention. With respect to the point stating “the author has presented an interesting paper in the "Opinion" section. I would recommend transforming it into a Review, since many of the author's statements are overly categorical and unsubstantiated. Also, mainly private facts are described that are not typical for all living organisms and their systems. We know of many cases where the holobiont concept does not work, for example, when obtaining virus-free plants from cell cultures.” I have included additional information following your recommendations that I think have improved the manuscript to the extend of being considered as a Review, and I have tried to avoid being categorical as well as giving additional examples. The virus-free plants from cultures have being discussed in the text as I believe it can be very useful in the context of the manuscript. Once again, I thank this reviewer for helping in improving this manuscript that I hope can be considered now for publication. |
||

Reviewer 5 Report
Comments and Suggestions for Authors
In this mini-review, the author describe as the holobiont concept has changed our understanding in ecology under diverse studies in host-microbial interaction with important impact in evolution, and with a perspective on stress biology for the comprehension of diverse mechanism with a only propose, adaptation or survivor during colonization. The minireview is very interesting, the information is well-performed and well-written.
I suggest the following revisions to further improve this minireview.
Here is my observation:
1.-In introduction, the lines 71-76 most be edited
2.- Figures 1 and 2, please describe in text.
3.-In lines 196-198; describe these mechanisms or contributions mentioned for the microbiome at the anthosphere.
4.-In lines 189-196, some references are necessary for these interactions.
5.-In text, the mention and description for table 1 is missing.
6.-I suggest edit the conclusions based on the objective of this mini-review.
Author Response
|
Comments 1: [In introduction, the lines 71-76 most be edited"]
|
|
|
Comments 2: [Figures 1 and 2, please describe in text.] |
|
Response 2: Agree. I have, accordingly, modified the text to describe the Figures 1 and 2 and an additional third figure that has been created. See lines 198; 465 and 598.
Comments 3: [In lines 196-198; describe these mechanisms or contributions mentioned for the microbiome at the anthosphere.]
Response 3: Thank you again for pointing this out. Based on your comment, I have expanded this paragraph for a better description of the interactions of the microbiome at the anthosphere.
Comment 4 [In lines 189-196, some references are necessary for these interactions.] Response 4: Indeed, you are right that it was necessary to add additional references to clarify these interactions. Such references are now included.
Comment 5 [In text, the mention and description for table 1 is missing.]
Response 5: Thank you for your suggestion. I have included two additional tables. The three Tables are mentioned in the text now. See lines 299; 431 and 630.
Comment 6 [I suggest edit the conclusions based on the objective of this mini-review.]
Response 6: Thank you for pointing this out. I have checked the conclusions section and modify them accordingly.
|
|
4. Response to Comments on the Quality of English Language |
|
Point 1: The English is fine and does not require any improvement. |
|
Response 1: I appreciate your comment in this respect. Nevertheless, an editing service has been hired, to improve the quality of English language. |

Round 2
Reviewer 1 Report
Comments and Suggestions for Authors
In the revision of the MS, the author improve the MS. But there are still some issues need to be corrected. Additionally, the format of the revision is not well-prepared. It takes my extra time to re-evaluate it.
Some points are listed as the followings:
- Title, suggest to change to ... roles of the holobionts...
- Line 28, the holobiont, add space.
- For figures, suggest to add some necessary annotations for the different elements in the diagram to make it more readable and easier to understand.
- For a review article, I think it’s better to supplement with 1-2 comprehensive diagrams that are easy to understand, and inspiring, thus will be very helpful for the readers.
- Line 1014, should Table 3, not Table 2. please recheck.
- There are still some format mistakes in the revision, please recheck them.
- Please recheck the abbreviations section, some abbreviations used in the MS are missing.
- The revised reference section is confusing, please recheck them for format issue.
Author Response
|
3. Point-by-point response to Comments and Suggestions for Authors |
|
Comments 1: [Title, suggest to change to ... roles of the holobionts...]
|
|
|
Comments 2: [Line 28, the holobiont, add space..] |
|
Response 2: I am sorry, but I have been unable to locate the lack of this space in line 28.
Comments 3: [For figures, suggest to add some necessary annotations for the different elements in the diagram to make it more readable and easier to understand.]
Response 3: Thank you again for pointing this out. Based on your comment, I have added annotations for the different elements in the diagram of Figure 1 and Figure 2. Hopefully they are easier to understand now.
Comment 4 [For a review article, I think it’s better to supplement with 1-2 comprehensive diagrams that are easy to understand, and inspiring, thus will be very helpful for the readers.] Response 4: Indeed, you are right that the inclusion of a diagram makes easier to understand and to inspire the reader. Therefore I have included Figure 4 to that end.
Comment 5 [For the two figures in the MS, the current content of the figures are relatively simple, and the information provided is limited. It is recommended to modify it and add more abundant contents.]
Response 5: Thank you for your suggestion. I think doubling the number of images improves the content, making it more complex and expanding the information. Thanks for the suggestion.
Comment 6 [Line 30, secondly, add a comma]
Response 6: Thank you for pointing this out. I have hired the MDPI editing service to review the whole article to avoid mistakes such as this one.
Comment 7 [Line 39, give the full name for SCFAs, when it appears for the first time in the MS.]
Response 7: Previous response also applies here.
Comment 8 [Line 76, no period at the end of the sentence. Please check.]
Again, previous response also applies here. Thank you again.
Comment 9 [Line 220, check the title for format inconsistency.]
Response 9: Thank you very much. The title has been modified to avoid formatting inconsistencies.
Comment 10 [Line 315, VOCs, not need to give the full name here.]
Response 10: Thank you very much, the error has now been corrected.
Comment 11 [References: please correct many format errors. Also, the number of the references is not enough. I suggest to add more influential articles. The current 63 reference seems not enough for covering this important topic.]
Response 11: I appreciate the reviewer’s comment and have checked for format errors. The former number of 63 references have been increased.
Comment 12 [Table 1. please recheck the format of the table. Also, the information in this table is limited. Please revise and expand this table.]
Response 12: I thank the reviewer for raising this issue. As previously mentioned, and in order to deal with the expansion two additional tables (Table 1 and Table 2) have been included.
|
|
4. Response to Comments on the Quality of English Language |
|
Point 1: The English could be improved to more clearly express the research. |
|
Response 1: I appreciate your comment in this respect. The editing service has been hired, to improve the quality of English language. |

Reviewer 3 Report
Comments and Suggestions for Authors
Authors addressed most of the comments. Yet, the language needs polishing. Moreover, some parts of the manuscript have been copied from other sources (without rephrasing).
Figures have been wrongly placed and numbered wrongly. Authors should refer the figures in the text, as well.
Author Response
|
3. Point-by-point response to Comments and Suggestions for Authors |
|
Comments 1: [Authors addressed most of the comments. Yet, the language needs polishing. Moreover, some parts of the manuscript have been copied from other sources (without rephrasing).]
|
|
Response 1: Thank you for acknowledging that your previous comments have been addressed. With respect to the language, the MDPI editing and translation service has been used to improve the use of English.
|
|
Comments 2: [Figures have been wrongly placed and numbered wrongly. Authors should refer the figures in the text, as well.] |
|
Response 2: Thank you again for pointing this out. I have replaced the figures and correctly numbered. Also, figures are referred in the text now.
|
Once again, thank you for helping in this respect.

Reviewer 4 Report
Comments and Suggestions for Authors
The author has answered the questions posed by the reviewer in a meaningful way and changed the manuscript in accordance with the wishes. It seems to me that in this form, despite the non-standard ideas put forward by the author, the manuscript can be published.
Author Response
|
1. Summary |
|
|
||
|
Once again, thank you very much for taking the time to review this manuscript. We appreciate that you consider our manuscript as an interesting paper and for considering it for publication
|
||||
|
2. Questions for General Evaluation |
Reviewer’s Evaluation |
Response and Revisions |
||
|
Does the introduction provide sufficient background and include all relevant references? |
Yes |
Thank you.
|
||
|
Are all the cited references relevant to the research?
|
Yes |
|
||
|
Is the research design appropriate?
|
Yes |
|
||
|
Can be improved |
|
||
|
Can be improved |
|
||
|
Are all figures and tables clear and well-presented? |
Yes |
|
||
|
3. Point-by-point response to Comments and Suggestions for Authors |
||||
|
Comments 1: [The author has answered the questions posed by the reviewer in a meaningful way and changed the manuscript in accordance with the wishes. It seems to me that in this form, despite the non-standard ideas put forward by the author, the manuscript can be published.]
|
||||
|
Response 1: Thank you for acknowledging the modifications and for considering that the manuscript can be published now.
|
||||
